# A High-Throughput Inhibitor Screen Targeting CLAG3 Export and Membrane Insertion on Human Erythrocytes Infected with Malaria Parasites

**DOI:** 10.3390/pathogens14060520

**Published:** 2025-05-23

**Authors:** Jinfeng Shao, Jonathan Chu, Kashif Mohammad, Sanjay A. Desai

**Affiliations:** Laboratory of Malaria and Vector Research, National Institute of Allergy and Infectious Diseases, National Institutes of Health, Rockville, MD 20852, USA; jinf.shao@outlook.com (J.S.); jonathan.chu627@gmail.com (J.C.); mohammad.kashif@nih.gov (K.M.)

**Keywords:** *Plasmodium falciparum*, malaria, high-throughput screening, CLAG3, nutrient uptake, ion channel, antimalarial drug discovery

## Abstract

To facilitate intracellular growth and replication, the virulent human malaria parasite *P. falciparum* remodels its host erythrocyte by exporting many proteins into the host cell cytosol. Along with a few other exported proteins, the parasite CLAG3 protein is then inserted in the host erythrocyte membrane, exposing a small variant loop to host plasma and contributing to essential nutrient acquisition via the plasmodial surface anion channel (PSAC). To explore trafficking mechanisms and develop therapies that block host cell remodeling, we have now used a split NanoLuc reporter and performed a high-throughput screen for inhibitors of parasite CLAG3 trafficking and insertion at the host membrane. We screened ~52,000 small molecules and uncovered 65 chemically diverse hits. Hits that inhibit the NanoLuc reporter without blocking protein export were filtered out by a secondary screen whose signal does not depend on protein export. Because chemicals that interfere with parasite maturation were found to compromise CLAG3 export indirectly, a third screen using a NanoLuc reporter-tagged intracellular protein was used to evaluate nonspecific toxicity. Although our relatively small chemical screen did not identify bona fide inhibitors of CLAG3 host membrane insertion, these studies establish a framework for larger screens to identify novel export inhibitors. Such novel inhibitors will provide important insights into how *Plasmodia* remodel their host cells and may seed the development of therapies that block the export and membrane insertion of proteins needed for intracellular parasite survival.

## 1. Introduction

Malaria remains a leading infectious cause of morbidity and mortality worldwide. Acquired resistance to available antimalarial drugs [1], compromised vector control due to insecticide resistance [2], and limited efficacy of two approved malaria vaccines [3] have all hindered attempts at malaria control and thwarted the goal of malaria elimination in endemic countries. Climate change, human and vector migration, war and civil unrest, and the COVID-19 pandemic have interfered with control efforts and led to increases in global malaria cases in the last decade. New antimalarial drugs that block unexploited parasite targets are needed.

*Plasmodium falciparum*, the most virulent human malaria parasite species, exports many proteins into its host cell cytoplasm, dramatically remodeling the infected cell to facilitate intracellular parasite growth and replication [4]. Several of these exported proteins insert in the erythrocyte membrane and are therefore exposed to host plasma, where they serve diverse roles in cytoadherence [5], immune evasion [6,7], virulence [8], and nutrient acquisition. While most surface-exposed antigens are not conserved in the genus *Plasmodium*, CLAG3 is conserved in all examined malaria parasite species that infect primates, rodents, and birds; this surface-exposed protein is encoded by members of the *clag* multigene family [9,10,11].

A fundamental consequence of host cell remodeling is the marked increase in erythrocyte membrane permeability to organic and inorganic solutes. Single channel and whole-cell patch-clamp of infected erythrocytes established that a single broad selectivity ion channel termed the plasmodial surface anion channel (PSAC) is responsible for this permeability [12,13]. Subsequent genetic mapping and DNA transfection experiments linked PSAC activity to CLAG3 [14], the 167 kDa protein conserved in all *Plasmodium* spp. and previously assumed to function in either host cell invasion or cytoadherence [15,16]. Its role in nutrient uptake was unexpected because CLAG3 lacks homology to known ion channel proteins in other organisms [17,18]. Consistent with a direct role in PSAC formation, CLAG3 is integral at the host membrane [19,20]. Knockdown and knockout studies suggest it oligomerizes after membrane insertion to form the nutrient channel pore [18,21]. CLAG3 forms a ternary RhopH complex with RhopH2 and RhopH3 [20,22,23,24], two unrelated proteins. Paralleling conservation of PSAC activity on infected mammalian and bird erythrocytes [25], the *clag*, *rhoph2*, and *rhoph3* genes are all strictly conserved in *Plasmodium* spp. [11,26].

In contrast to what might be expected for channel-forming proteins, the three subunits of the RhopH complex are manufactured as soluble proteins. Also surprisingly, they are made in the preceding parasite cycle, packaged as a complex in rhoptries, and transferred to new erythrocytes during host cell invasion [20,27,28,29]. This transfer deposits the soluble complex into the parasitophorous vacuole surrounding the growing parasite. Although initially debated [30], the complex is now thought to interact with the PTEX export translocon to be transferred into host cell cytosol [20,22]. While other exported proteins must be unfolded for transit through the translocon and then refolded in erythrocyte cytosol by parasite-encoded chaperones [31,32], it is not clear that the large ternary RhopH complex would tolerate disassembly and reassembly. Either during or shortly after PTEX interaction, CLAG3 becomes integral to membranes, as evidenced by alkaline extraction experiments [22]. Finally, after a brief transit via the Maurer’s clefts [33], the complex is inserted into the host erythrocyte membrane to form the nutrient channel pore directly or indirectly.

As the many unusual aspects of RhopH complex trafficking and membrane insertion remain poorly understood, we have now leveraged a recently developed reporter of CLAG3 host membrane insertion and surface exposure to screen for trafficking inhibitors [34]. The engineered reporter utilizes a HiBiT fragment of NanoLuc inserted into a small extracellular loop on CLAG3. CLAG3 insertion at the host membrane can then be tracked by complementation with LgBiT, a large membrane-impermeant NanoLuc fragment, to produce a luminescent signal proportional to the amount of surface-exposed CLAG3 at the host membrane. We have miniaturized and optimized this assay for a 384-well microplate format and executed a CLAG3 export inhibitor screen. Hits were filtered to exclude two types of false positive hits. The results are considered with the goal of discovering antimalarial lead compounds that interfere with the formation of the conserved and essential PSAC nutrient acquisition pathway. In addition to providing starting points for antimalarial drugs, positives from these screens will be important reagents for the study of parasite protein export and membrane insertion on cells infected with *Plasmodium* spp. and other intracellular pathogens.

## 2. Materials and Methods

### 2.1. In Vitro Cultivation of Plasmodium falciparum Strains

The wildtype KC5 clone and its transfected derivatives were cultivated in O^+^ human erythrocytes obtained from commercial sources (University of Virginia Blood Bank, Charlottesville, VA, USA) at 5% hematocrit in standard media—RPMI 1640 supplemented with 28.6 mM NaHCO_3_, 25 mM HEPES, 50 µg/mL hypoxanthine, 10 mg/L gentamicin, and 0.5% Albumax I lipid-rich BSA (Thermo Fisher Scientific, Waltham, MA, USA). Parasites were cultivated at 37 °C under 90% N_2_, 5% CO_2_, and 5% O_2_. The *8-1-3HA* integrant line was maintained without selection while *GBP-HB*, which expresses a 99-residue leader sequence from Pf3D7_1016300 followed by a cassette containing 5 HiBiT tags and an HA epitope tag, was maintained with 1.5 µM DSM1 for the yDHODH selection marker.

### 2.2. Miniaturized Luminescence Measurements Using Split NanoLuc

The *8-1-3HA* clone was used for a high-throughput inhibitor screen targeting CLAG3 export and surface exposure. In total, 52,480 compounds were examined in individual wells using a 384-well microplate format. Compounds from available chemical libraries were dispensed by 100 nL pin transfer of a 10 mM DMSO stock into microplate wells preloaded with 20 µL of phenol-free medium (Gibco phenol red-free RPMI 1640 supplemented with 50 µg/L hypoxanthine and 10 mg/L gentamicin but without Albumax I). After plate vortex and centrifugation, 40 µL of a cell culture resuspension containing 3–5% synchronized ring-stage parasites in the same phenol-free medium with 0.5% Albumax I was added using a microplate dispenser (Matrix Wellmate, Thermo Fisher Scientific) to yield a screening hematocrit of 0.1% and a screening compound concentration of 16.7 µM. Negative control wells contained 100 nM artemisinin (ART). After mixing, plates were sealed with Breathe-Easy sealing membrane (Research Products Int., Mt Prospect, IL, USA) and incubated at 37 °C under 5% CO_2_ for 24 h. After subsequent removal of the sealing membrane, 50 µL of the culture supernatant was aspirated (BioTek EL406 microplate washer, Winooski, VT, USA) before addition of 20 µL ice-cold buffer containing a 100:1:2:1 ratio mixture of Nano-Glo HiBiT Extracellular Buffer (Promega, Madison, WI, USA)–LgBit protein–Extracellular furimazine Substrate–5% Albumax I in dH_2_O. After mixing and a 30 min room temperature incubation in the dark, luminescence was measured using the Centro XS3 LB960 reader (*GBP-HB*) with a counting time of 0.5 s/well.

The counterscreen for false positive hits that inhibit NanoLuc-mediated luminescence without affecting CLAG3 export followed the above protocol except that synchronized ring-stage cultures were cultivated for 24 h to allow maturation to the trophozoite stage prior to plating with compounds. After resuspension in 384-well microplates, the cell suspension was allowed to settle at room temperature before aspiration (BioTek EL406 microplate washer) and addition of the same Nano-Glo HiBiT Extracellular Buffer mixture and luminescence measurement as above.

The toxicity screen using *GBP-HB* parasites was performed using synchronized ring-stage cultures as performed for the CLAG3 export inhibitor screen except that luminescence measurement utilized the Nano-Glo Lytic Detection System (Promega) with a 100:1:2 ratio mixture of Nano-Glo HiBiT Lytic Buffer–LgBit protein–Extracellular furimazine Substrate without Albumax I supplementation.

For all three screens, in-plate DMSO and ART controls were normalized to 100% and 0% with linear interpolation of all compound-containing wells.

Luminescence measurements in secondary studies with chemical hits obtained in resupply (ChemBridge, San Diego, CA, USA) were performed using identical procedures with in-plate DMSO and ART control wells. The 50% effective concentrations for hits (*EC*_50_ values) were determined from dose–response experiments using 4× serial dilutions from the screening concentration.

### 2.3. Growth Inhibition Studies

Parasite killing by screening hits was evaluated using an SYBR Green I-based fluorescence assay for parasite nucleic acid in a 96-well format as previously described [35]. Ring-stage cultures were seeded with inhibitors after sorbitol synchronization at 0.8% parasitemia and 2.5% hematocrit in the standard RPMI 1640 culture medium or in the modified PGIM culture medium [35], both with 0.5% Albumax I lipid-rich BSA supplementation. Cultures were maintained for 72 h at 37 °C under standard culture conditions before the addition of 100 µL lysis buffer containing SYBR Green I at a 2X concentration (Thermo Fisher Scientific) and 20 mM Tris, 10 mM EDTA, 1.6% triton X100, 0.016% saponin at pH 7.5. After a 60 min incubation in the dark, parasite nucleic acid production was quantified with fluorescence measurements (excitation/emission of 485 nm/528 nm). Inhibitor *IC*_50_ values were estimated from triplicate measurements at seven serial-dilution drug concentrations; measurements were normalized to DMSO controls after subtraction of background fluorescence from matched cultures growth with 20 μM chloroquine.

### 2.4. Immunofluorescence Assays

Indirect immunofluorescence assays were performed using thin smears on microscope slides after air-drying and fixation with 1:1 acetone–methanol at −20 °C. Slides were then dried and blocked with 3% milk in PBS for 45 min at RT before incubation with primary antibodies in the same blocking buffer (mouse anti-HA at 1:100 dilution; rabbit anti-RhopH3, 1:500) for 2 h at RT under coverslips. Slides were then washed twice with ice-cold PBS for 10 min each before the addition of Alexa Fluor 488-conjugated anti-mouse or 594-conjugated anti-rabbit secondary antibodies at 1:500 dilution with 10 µg/mL 4′, 6-diamidino-2-phenylindole (DAPI) in blocking buffer. After a 45 min RT incubation, slides were washed twice with ice-cold PBS, dried, and mounted with Prolong Diamond antifade mountant (Molecular Probes, Eugene, OR, USA). Images were captured on a Leica SP8 confocal microscope with 405, 488, and 594 nm excitation and a 64 X oil objective. With each slide, at least 10 fields with more than 100 cells each were scanned to obtain representative images for processing using Leica LAS X and Fiji ImageJ 1.53 software.

## 3. Results

### 3.1. An Optimized 384-Well Microplate Assay for CLAG3 Exposure on Infected Cells and a High-Throughput Screen for Inhibitors

We began these studies with the Reporter of Insertion and Surface Exposure (RISE), a cell-based assay recently developed to track surface exposure of CLAG3 protein on infected human erythrocytes. We selected the *8-1-3HA* parasite engineered to express the HiBiT fragment of NanoLuc followed by a 3× HA epitope tag within the CLAG3 hypervariable domain (HVR), which is exposed on mature infected cells (Figure 1A). HVR replacement with the 27-residue reporter sequence did not compromise CLAG3 host membrane insertion, had no effect on PSAC-mediated solute permeabilities, and permitted tracking of CLAG3 export via a bright luminescence signal resulting from binding with extracellular LgBit [34].

Using a miniaturized 384-well microplate format, 100 nM artemisinin (ART) and DMSO only were chosen as positive and negative controls for signal inhibition because we found that ART produces rapid parasite killing and prevents luminescence signal development. We optimized parasite stage, parasitemia, hematocrit, buffer conditions, volume and compound addition conditions, incubation at 37 °C to allow parasite maturation and CLAG3 export, and subsequent processing of plates to define the optimal signal-to-background for screening. The optimized cell-based assay utilized a 24 h incubation of ring-stage infected cells with test compounds and achieved reproducible differences between positive and negative controls in a small 60 µL volume with a low 0.1% hematocrit (Figure 1B), enabling large-scale chemical screens with modest investments in parasite culture, reagents, and labor. A signal-to-background ratio of 5.9 indicated an excellent window for inhibitor identification. We also used the Z’ factor, a statistical indicator of assay reliability for high-throughput screening [36]; this statistic yields a value of 1.0 for a perfect noise-free assay and predicts a false-positive rate of <1% with values above 0.5. Our cell-based luminescence reporter assay produced a Z’ of 0.60 ± 0.03 (Figure 1C), indicating a low-noise assay well suited for high-throughput inhibitor screens.

We therefore executed a >52,000 compound screen for inhibitors of CLAG3 export and host membrane insertion; compounds were screened at a relatively high concentration, 16.7 µM estimated from compounding pinning conditions and our optimized screening volumes, to allow detection of relatively weak inhibitors. In-plate ART positive and DMSO negative controls were used to normalize readings to 0 and 100% CLAG3 membrane insertion (red and blue histogram bars, respectively, Figure 1D). The normalized results for wells containing single compounds from our chemical libraries revealed that most compounds were without effect on CLAG3 export (Cpds, dark yellow histogram bars, Figure 1D), but identified 177 small molecules that reduced the reporter signal by at least 50%. The resulting hit rate, 0.3%, is within the broadly accepted hit rates for chemical screens against diverse targets [37].

### 3.2. Hits Filtered for False Positives That Inhibit NanoLuc Signal Development

Although the split NanoLuc-based RISE assay reports specifically on CLAG3 insertion at the host membrane, we recognized that a subset of hits in our screens may reduce the luminescence signal by inhibiting NanoLuc directly without affecting CLAG3 export. Such false positives may work by blocking LgBit-HiBit association or inhibiting furimazine oxidation to furimamide, the chemical reaction that yields the light signal. NanoLuc structure-function studies support this concern as they reveal a surface binding site that can inhibit substrate access through negative allostery [38]. We therefore devised a cell-based counterscreen to filter our screening results for false positives that act only on the NanoLuc reporter.

In contrast to the immature ring-stage cells used in the primary screen, our counterscreen used synchronized trophozoite-stage *8-1-3HA* infected cells. We reasoned that these mature cells already display surface CLAG3 and should produce bright luminescence in our assay in the absence of NanoLuc inhibition. Instead of the 24 h incubation of immature infected cells with library compounds in the primary screen, this counterscreen required only a brief incubation with mature infected cells. To permit normalization, we used matched cultures treated with 150 nM artemisinin for 24 h as in-plate negative controls. When tested in parallel with the trophozoite-stage infected cells, these controls, seeded in columns 23 and 24 of each 384-well plate (red symbols, Figure 2A), predict the NanoLuc block. A positive control, with no compound addition in columns 1 and 2 (blue symbols), permitted normalization to 100% for compounds without effect on the NanoLuc reporter.

Our counterscreen evaluated 16,320 compounds, covering all primary screen mother plates yielding one or more hits in the CLAG3 export assay. It identified 17 hits that block the NanoLuc reporter by ≥50% (Figure 2B). The counterscreen’s lower hit rate is consistent with the expectation that these false positives represent a subset of hits from the primary screen.

Figure 2C shows the results from the NanoLuc counterscreen plotted against the primary screen for all compounds. Hits that directly block NanoLuc signal development without affecting CLAG3 export or parasite maturation clustered along the diagonal (blue dashed line) while those that do not inhibit NanoLuc have preserved signals in the counterscreen (y = 100%, red line). Notably, this plot reveals that nearly all hits are segregated into these two distinct groups, further confirming the reproducibility of our single measurement assay and high-throughput screening results.

### 3.3. Secondary Studies

#### 3.3.1. Excellent Retest Rate for Hits Obtained in Resupply

Based on results from the two screens described above, we selected 38 hits with potent blocking activity in the primary screen and negligible activity in the NanoLuc counterscreen; 33 were available for resupply. Secondary studies with these resupplied hits confirmed activity for 19 compounds, yielding a 58% retest rate for this cell-based miniaturized assay for CLAG3 export and membrane insertion. We noticed that the many compounds that did not reproduce were in the last microplate column having compounds from the screening library, suggesting rare but detectable edge effects that appeared to result from nonuniform dispensing of cell suspensions or screening compounds. Some compounds also might not have reproduced due to chemical degradation or other issues with vendor compound resupply.

The structures of reproducible hits and their activities in the primary and NanoLuc-block screens are shown in Table 1. Examination of these structures revealed a diverse collection of compounds without significant clustering of scaffolds. PEI-14 and PEI-27 had the greatest pairwise similarity with a Tanimoto similarity coefficient of 0.75; similarity was lower for all other pairwise comparisons, with PEI-5 and PEI-23 (Tanimoto score of 0.48) being the next highest score [39].

We then compared the results of the secondary studies to the primary screen result and observed an excellent correlation for these 19 compounds, validating our screening strategy and supporting assay reproducibility (Figure 3A). Dose–response studies revealed that effective concentrations required to reduce the luminescence signal in the CLAG export assay (*EC*_50_ values) were generally in the low micromolar range with several compounds having submicromolar potency (Figure 3B).

#### 3.3.2. Localization and Growth Inhibition Studies Suggest Off-Target Effects

Our screen predicts that hits will either interfere with CLAG3 trafficking to or insertion in the host membrane. Thus, we next used indirect immunofluorescence assays (IFAs) with confocal microscopy to examine CLAG3 trafficking (Figure 4). These experiments used erythrocytes infected with ring-stage *8-1-3HA* parasites and cultivation with individual hit compounds for 24 h to follow our primary screen’s conditions. CLAG3 localization was then visualized with an anti-HA antibody specific to the introduced epitope tag [34]. Tandem labeling with anti-RhopH3 permitted tracking of a second PSAC component; depending on the precise mechanism of inhibitor action, this protein’s trafficking may or may not be compromised. While IFA with control DMSO treatment revealed normal export, artemisinin treatment yielded reduced export of both CLAG3 and RhopH3 and more intense staining of smaller intracellular parasites, consistent with parasite killing by artemisinin. IFA was then performed with all 19 compounds, with several of the most potent compounds shown in Figure 4. With each of these hits, we were unable to detect changes in CLAG3 or RhopH3 distribution. Imaging, however, suggested compromised parasite maturation, as indicated by smaller parasites with retained intracellular probe labeling, as exemplified by the images shown for PEI-6, PEI-10, and PEI-27. While prior studies with these antibodies have established their specificities for the corresponding epitopes [21,34], the associated IFA fluorescence signals may exhibit nonlinear dose–response intensities and prevent unambiguous detection of the reduced CLAG3 export.

Because we recognized that these factors may prevent clear identification of one or more true hits from our screen, we sought an independent method to evaluate reduced CLAG3 export. An important phenotype associated with either CLAG3 knockout or failed delivery and host membrane insertion is compromised growth in PGIM, a medium with reduced but more physiological concentrations of key nutrients than the standard RPMI 1640-based formulation used in routine *P. falciparum* cultivation. We predicted that inhibitors of CLAG3 insertion at the host membrane would be more effective in parasite growth inhibition assays (GIAs) using PGIM- than RPMI 1640-based media. Notably, PSAC inhibitors are significantly more effective in PGIM with 10 to 800-fold lower *IC*_50_ values than obtained in dose–response studies using RPMI 1640 [35]. We therefore performed dose–response GIA experiments and found unchanged or modestly weaker activities in experiments using PGIM when compared to those using RPMI 1640 (Figure 5). This result contrasts with the expected improvement in parasite killing under nutrient restriction; it again suggests that these screening hits compromise parasite growth by mechanisms other than blocked CLAG3 host membrane insertion.

#### 3.3.3. A Separate Reporter Line for Excluding Nonspecific Toxicity

Because the above studies suggested nonspecific parasite growth inhibition by the identified screening hits, we sought to distinguish direct inhibition of CLAG3 exposure on infected cells from indirect signal reductions due to nonspecific toxicity. Such toxicity may prevent parasite maturation and indirectly prevent CLAG3 surface exposure. We therefore used the *GBP-HB* parasite clone, which expresses a 99-amino acid GBP130 leader sequence followed by 5 copies of HiBiT (Figure 6A). The GBP130 leader sequence targets this soluble reporter protein to the host cytosol but, in contrast to CLAG3, the HiBiT sequence remains intracellular and is not exposed to the extracellular space. Use of the NanoLuc lytic buffer, which contains detergents to lyse infected cells, yields a luminescence signal by allowing interactions between the HiBiT tags and LgBit. Importantly, a reduced signal with this reporter line would implicate nonspecific toxicity due to reduced expression of the reporter protein. We miniaturized luminescence measurements with *GBP-HB* into 384-well microplate format, used in-plate DMSO and ART controls to normalize these readings, and compared the measurements to those obtained with *8-1-3HA*. With each screening hit, the block using the *GBP-HB* reporter matched or exceeded the inhibition seen with the *8-1-3HA* reporter for CLAG3 export (Figure 6B), establishing nonspecific toxicity as the primary mechanism of action.

Screening hits that interfere with CLAG3 export may also compromise parasite growth because failed CLAG3 export would prevent PSAC-mediated nutrient acquisition. Consistent with this, PSAC inhibitors and knockdown of channel components prevent parasite propagation and have validated PSAC as a drug target [20,40]. At the same time, two observations suggest that our CLAG3 export assay would not be compromised by indirect effects on parasite growth. First, our protocol tracks CLAG3 export and surface exposure during parasite maturation from the ring to the early trophozoite stage, a period in the intracellular cycle where PSAC activity is only minimally activated [41]. Although the channel subunits are synthesized in advance and delivered to new erythrocytes at the time of invasion, CLAG3 membrane insertion and PSAC-mediated nutrient uptake are not evident until >24 h later [34]. Thus, inhibitors of CLAG3 export and surface exposure are not expected to produce reduced *GBP-HB* reporter signals due to compromised parasite maturation. Second, although PSAC is essential, CLAG3 knockout does not compromise parasite growth in a standard RPMI 1640-based culture medium. The supraphysiological nutrient concentrations in this medium permit adequate nutrient uptake in CLAG3 knockout parasites because other CLAG paralogs can preserve PSAC activity at reduced levels [21,42]. Interestingly, CLAG3 knockout parasites do not grow in PGIM, a modified culture medium with reduced, more physiological levels of key nutrients acquired via PSAC [21,35]. Because our screens used a relatively short 24 h cultivation in RPMI 1640-based medium, we predict that CLAG3 export inhibitors will not compromise parasite maturation and would not yield reduced signals in the *GBP-HB* reporter assay.

To test this prediction, we used MBX-2366, a nanomolar active PSAC inhibitor that blocks the channel pore and kills parasites in standard 72 h growth inhibition experiments (also referred to as ISG-21, [35,40]). As MBX-2366 is highly specific with no action against a battery of transporters and unrelated enzymes [40], this inhibitor has advanced into the hit-to-lead pipeline with the identification of derivatives having suitable in vivo pharmacokinetics and ADME properties for antimalarial drug development. Here, we performed the *GBP-HB* assay with 200 nM MBX-2366, a concentration that effectively blocks nutrient uptake, and found negligible inhibition (Figure 6C). As the inhibitor may be adsorbed by the Albumax supplement added to the culture medium to provide essential serum lipids, we also performed the assay using an Albumax-free medium. The negligible inhibition of the *GBP-HB* reporter signal with MBX-2366 under both conditions indicates that loss of PSAC activity, either through inhibited CLAG3 export or through the block of the nutrient channel pore, does not interfere with parasite maturation or expression of the chimeric reporter protein inhibition.

## 4. Discussion

We report a screen for small molecule inhibitors of malaria parasite protein export and surface membrane insertion on infected erythrocytes. Our strategy utilized the split NanoLuc technology with the insertion of a small 11 aa HiBiT fragment into an extracellular loop of CLAG3, a strictly conserved protein in *Plasmodia* that determines channel-mediated nutrient uptake. The addition of LgBit complements the exposed HiBiT to restore NanoLuc activity and produce luminescence, quantitatively tracking CLAG3 delivery and insertion at the erythrocyte surface. We miniaturized this reporter assay into 384-well microplates and screened a library of >52,000 diverse chemicals for inhibitors. Studies with inhibitors identified two separate mechanisms for false positive hits. First, chemicals that inhibit NanoLuc and indirectly reduce luminescence were identified and filtered by a modified assay using mature infected cells with CLAG3 already exposed on the host membrane. Notably, this counterscreen revealed two distinct categories of hits: chemicals whose blocking activity was fully attributed to NanoLuc inhibition (blue diagonal line, Figure 2) and those that worked primarily via inhibition of CLAG3 display (red horizontal line, Figure 2). A second mechanism for false positives, nonspecific toxicity that interfered with parasite maturation and indirectly hindered CLAG3 export, was identified using *GBP-HB*, an engineered line whose expression of a chimeric HiBiT reporter protein is reduced by toxic chemicals. Unfortunately, all hits selected for secondary studies exhibited this nonspecific toxicity.

Small molecule modulators of protein trafficking or of protein–protein interactions were once thought to be unidentifiable because workers assumed that such chemicals are too small to affect global effects on proteins with much higher molecular weights and large interaction surfaces with other proteins [43,44]. Advances in both our understanding of protein structure-function and in screening technologies have more recently led to important successes in identifying these interesting modulators. Small molecules that correct defective trafficking of the CFTR chloride channel may be the most well-known success, with drug combinations that simultaneously improve the delivery of CFTR to the plasma membrane and potentiate channel opening in routine clinical use for cystic fibrosis [45,46]. Despite multiple mutation haplotypes that produce distinct disease mechanisms, these drug combinations are now effective in 85% of cystic fibrosis patients [47]. There are other important examples, including compounds that alter protein translocation through endoplasmic reticulum translocons [48], repair mislocalization of autophagy proteins [49], or correct misfolded transporters to restore their delivery to target membranes [50].

In contrast to the above examples where chemical modulators repair defective protein trafficking, we sought small molecules that interfere with normal trafficking and membrane insertion of CLAG3, a *P. falciparum* protein that determines PSAC-mediated parasite nutrient acquisition [14]. Such inhibitors might have one or more of several distinct mechanisms of action. They could inhibit CLAG3 interaction with the PTEX translocon at the parasitophorous vacuolar membrane surrounding the intracellular parasite [51,52]. Although it is unclear whether CLAG3 is itself translocated through PTEX [30], PTEX knockdown prevents CLAG3 delivery and insertion at the host membrane [20]. The inhibitor might instead interfere with large-scale conformational changes required for the conversion of soluble CLAG3 into an integral membrane protein [22]. Alternatively, it could interfere with trafficking and processing at the Maurer’s clefts, a Golgi-like organelle produced in the host cytosol of infected erythrocytes [29,33], or with the poorly understood presentation of CLAG3 at the host cell surface [34]. Depending on which of these steps are affected, chemical modulators may either prove to be specific for CLAG3 trafficking or they may produce more global reductions in the export of parasite proteins and/or surface display of other parasite antigens such as PfEMP1 and RIFINs [5,8].

Notably, the split NanoLuc-based RISE assay can be applied to any surface-exposed parasite antigen as it requires only the introduction of a small HiBit tag into an exposed site on the specific parasite antigen of interest. We focused our study on CLAG3 because this antigen has an established role in essential nutrient acquisition and is highly conserved in *Plasmodium* spp. Inhibition of other antigen’s export may eventually prove to be more promising therapeutically.

We sought chemical modulators that interfere with CLAG3 presentation on infected cells based on several predictions. First, we predict that compromised CLAG3 delivery will reduce the formation of PSAC, a channel required for intracellular parasite maturation as the primary conduit for the uptake of key nutrients. Isoleucine, an essential amino acid that cannot be obtained from hemoglobin digestion and has low endogenous host membrane permeability [53,54], is a well-established nutrient requiring PSAC-mediated uptake [55], but there are likely others. The resulting reduction in PSAC activity is then predicted to compromise the in vivo growth of *Plasmodium* spp., suggesting its use as an antimalarial therapeutic. The most compelling evidence supporting this prediction is the production of CLAG3 knockout parasites, which exhibit unabated growth in nutrient-rich RPMI 1640-based media but fail to grow in more physiological PGIM [21,35]. Because these compounds interfere with PSAC formation, we also predict that they will be strongly synergistic in growth inhibition studies when combined with advanced pore-blockers of PSAC such as MBX-2366 [40]. Such combinations are expected to be synergistic because they combine the inhibited formation of channels on the infected cell surface with a block of channels that form. We also identified PEI-1 as a compound that rapidly kills ring-stage parasites to reduce CLAG3 export in our screen. Its *IC*_50_ in standard growth inhibition assays, 23 nM, suggests it should be advanced as an antimalarial drug lead, extending findings from prior studies with this chemical [56].

Although our screens did not identify clear hits meeting all our criteria, we successfully established simple methods with miniaturized assays to distinguish true hits from compounds that inhibit NanoLuc or produce nonspecific toxicity. We suspect that our screen of 52,000 compounds may have been too small to find the desired modulators. As discussed above, chemical modulators of protein–protein interactions or protein trafficking have generally required larger screens with the examination of large numbers of screening hit derivatives [45]. Our facile workflow with miniaturization of both the primary screen and subsequent counterscreens to exclude nonspecific inhibitors will enable the required larger screens. As the envisioned chemical modulators will be important tools for understanding parasite biology and may lead to the development of highly synergistic antimalarial drug combinations, these larger screens are warranted.

## Figures and Tables

**Figure 1 pathogens-14-00520-f001:**
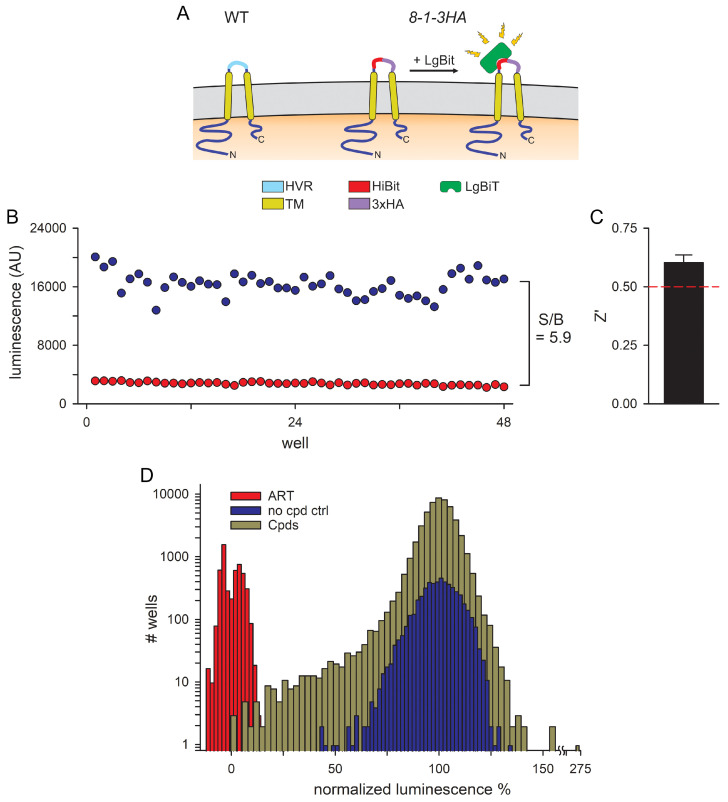
A miniaturized cell-based screen for inhibitors of CLAG3 export. (**A**) Schematic showing the wildtype CLAG3 topology in the host erythrocyte membrane (left, WT) and the engineered *8-1-3HA* parasite, which replaces the surface-exposed hypervariable region (HVR) with HiBiT and 3× HA tags. Extracellular addition of LgBit permits interaction with HiBiT to produce bioluminescence. (**B**) Luminescence measurements in 384-well microplate format using ring-stage *8-1-3HA* parasites grown without and with artemisinin (blue and red circles, respectively) for 24 h. The signal/background ratio of 5.9 establishes clear separation of positive and negative control wells. (**C**) Mean ± S.E.M. Z’ statistic from experiments as in panel (**B**); *n* = 3 independent trials. A Z’ value ≥ 0.5 (red dashed line) indicates a high confirmation rate in secondary studies [36]. (**D**) All-points histogram of screened compounds (gold) normalized to in-plate no compound and artemisinin controls (blue and red, respectively). Results are shown on a log-scale histogram to highlight hits, defined as compounds producing normalized luminescence values ≤ 50%. Three compounds producing increased luminescence values > 150% were not evaluated further; prior studies suggest such agonists are false positives that interact with NanoLuc to produce increased luminescence.

**Figure 2 pathogens-14-00520-f002:**
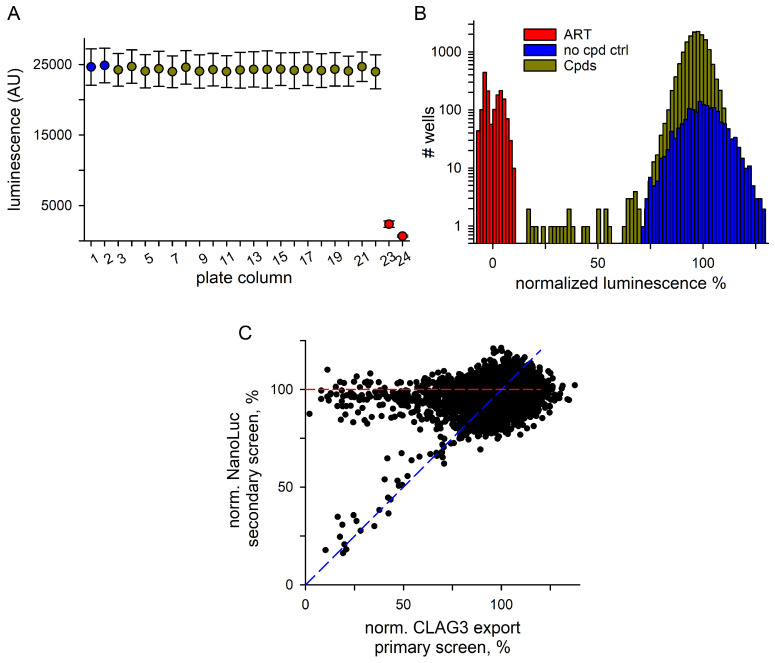
A second screen to exclude hits that inhibit NanoLuc. (**A**) Mean ± S.D. luminescence readings for each column of plates screened for NanoLuc inhibition. Values were calculated from all plates tested on one screening day with in-plate no compound and artemisinin controls (blue symbols for columns 1–2 and red for columns 23–24, respectively) and library compounds (gold, columns 2–21). (**B**) All-points histogram of normalized luminescence values for the NanoLuc inhibition screen. Fewer inhibitors were found in this screen. (**C**) Normalized block in NanoLuc inhibition screen plotted against a normalized block in the CLAG3 export primary screen, with each circle representing a single compound from the library; *n* = 16,320 compounds shown. Hits that block CLAG3 export without inhibiting NanoLuc cluster along the red dashed line, while inhibitors acting against NanoLuc only fall along the *y* = *x* diagonal (blue dashed line).

**Figure 3 pathogens-14-00520-f003:**
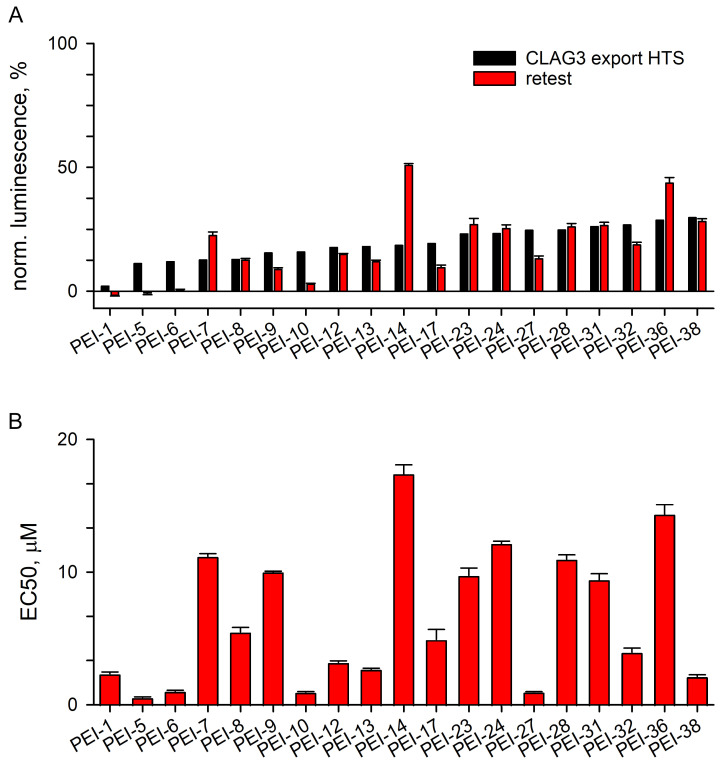
Retest and potency studies with identified hits. (**A**) Normalized block in the CLAG3 export assay with indicated hits. The primary screen block value (black bars) is shown with mean ± S.E.M. value from retest studies (red, *n* = 3 trials), obtained through resupply. Note the good correlation between primary and retest block levels. (**B**) Mean ± S.E.M. effective concentration producing 50% reduction in luminescence (*EC*_50_) for each hit, determined from dose–response experiments using the CLAG3 export assay. Most hits exhibit low micromolar potency in this 24 h assay.

**Figure 4 pathogens-14-00520-f004:**
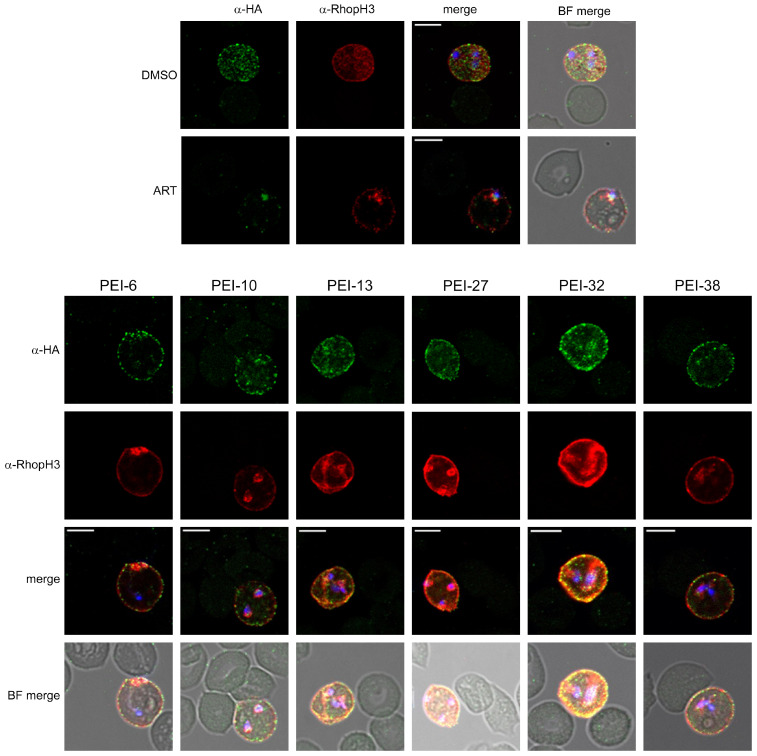
Indirect immunofluorescence assays (IFAs) with confocal microscopy to evaluate CLAG3 export block. Indicated antibodies and DAPI staining of parasite nucleic acid are shown for DMSO and artemisinin controls (**top**) and selected hits (**bottom**). The anti-HA and anti-RhopH3 antibodies probe localization of CLAG3 and the PSAC-associated RhopH3 proteins, respectively. Scale bars, 5 µm.

**Figure 5 pathogens-14-00520-f005:**
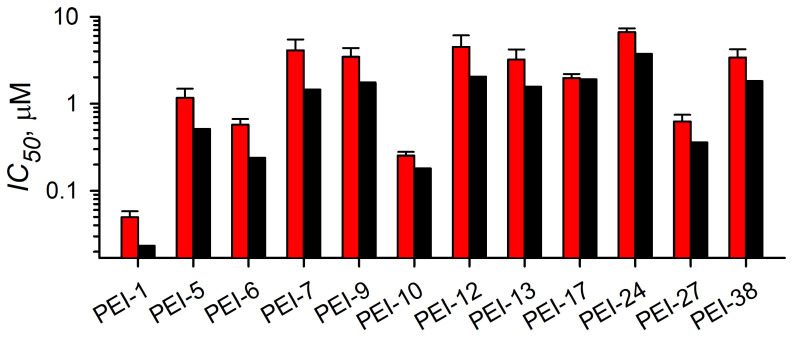
Mean ± S.E.M. parasite growth inhibition *IC*_50_ values in SyBR Green assays using 72 h propagation with indicated screening hits. Assays were performed using standard RPMI 1640-based medium or PGIM, a modified medium with reduced, more physiological concentrations of key essential nutrients acquired via PSAC (black and red bars, respectively). *IC*_50_ values were determined from dose–response studies.

**Figure 6 pathogens-14-00520-f006:**
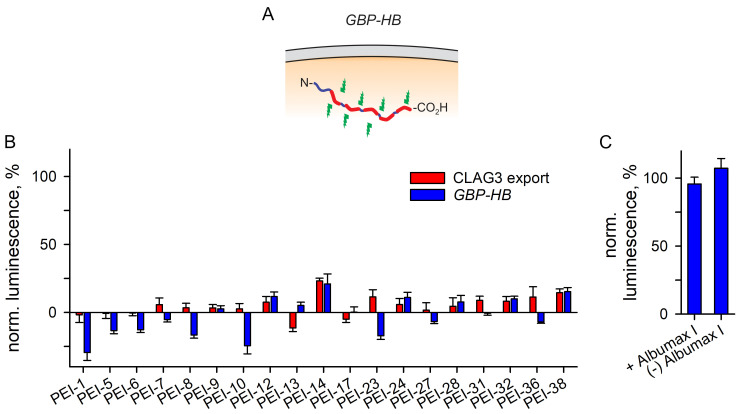
The *GBP-HB* reporter assay for nonspecific toxicity. (**A**) Schematic showing the engineered *GBP-HB* parasite, which expresses a chimeric reporter protein consisting of a 99 residue N-terminal leader GBP-130 sequence for export to erythrocyte cytosol followed by 5 copies of the HiBiT tag. Use of NanoLuc lytic buffer with LgBit produces bioluminescence (green lightning bolts). (**B**) Mean ± S.E.M. luminescence with indicated screening hits, added at screening concentrations, normalized to 100% and 0% for in-plate DMSO only and ART controls, respectively. Experiments were identically performed with *8-1-3HA* and *GBP-HB* reporter parasites (red and blue bars, respectively). *n* = 3 independent trials for each inhibitor and reporter parasite. Note, equivalent or greater inhibition using *GBP-HB*, indicating that each hit interferes with parasite maturation to compromise reporter protein expression. (**C**) Mean ± S.E.M. luminescence in *GBP-HB* with 200 nM MBX-2366, a potent and specific PSAC pore-blocker using a 24 h incubation in standard medium with or without Albumax I supplementation (left and right bars). PSAC inhibitors are not toxic to ring-stage parasites as the channel is activated at a more mature trophozoite stage.

**Table 1 pathogens-14-00520-t001:** Chemical structures and activities of hits. Values represent normalized luminescence for each compound, as measured in the indicated high-throughput screen (HTS).

ID	Structure	CLAG3 Export HTS, %	NanoLuc HTS, %
PEI-1, UCF-501	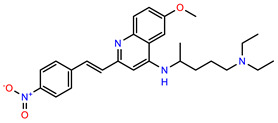	2.0	87.4
PEI-5	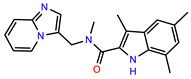	11.2	110.1
PEI-6	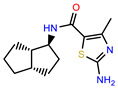	11.9	94.3
PEI-7	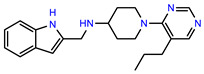	12.6	97.3
PEI-8	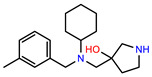	12.8	98.0
PEI-9	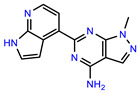	15.4	101.3
PEI-10	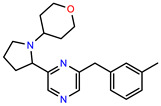	15.8	97.4
PEI-12	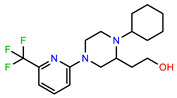	17.6	103.8
PEI-13	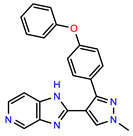	18.0	84.4
PEI-14	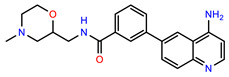	18.5	93.9
PEI-17	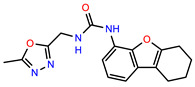	19.2	96.4
PEI-23	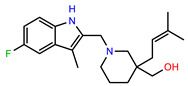	23.1	98.4
PEI-24	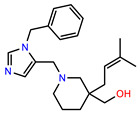	23.3	96.1
PEI-27	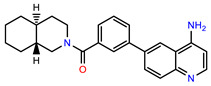	24.6	96.9
PEI-28	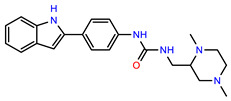	24.8	103.9
PEI-31	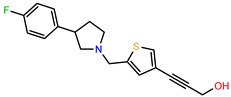	26.1	106.6
PEI-32	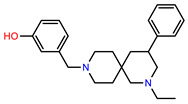	26.8	88.1
PEI-36	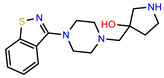	28.7	97.0
PEI-38	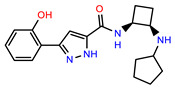	29.7	94.8

## Data Availability

All data is contained within the article. Further inquiries can be directed to the corresponding author.

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
