# Peer review of "A High-Throughput Inhibitor Screen Targeting CLAG3 Export and Membrane Insertion on Human Erythrocytes Infected with Malaria Parasites"

_pathogens, 2025, doi:10.3390/pathogens14060520_

Round 1
Reviewer 1 Report
Comments and Suggestions for Authors
Review of “A high-throughput inhibitor screen targeting CLAG3 export and membrane insertion on human erythrocytes infected with malaria parasites” by Shao et al for Pathogens.
Comments.
- This is an interesting screen where the authors tried to find compounds that blocked the display of CLAG3 on the surface of the parasite infected RBCs. The screen worked by incorporating the HiBit region of Nluc into the region of CLAG displayed on the surface and then using LgBit of Nluc to produce a bioluminescent signal if CLAG3 was displayed. A reduction of signal indicated less CLAG3 display however follow up screens found that the hit compounds where either blocking Nluc activity or reducing the growth of the parasites. In the end no specific inhibitors of CLAG3 exposure on the RBC surface were found but the approach appeared to be robust.
- Line 225. It says “yellow bars, Fig. 1D” but I can’t see yellow bars in 2D.
- In Fig 1D, the signal in some wells (bronze columns) is much stronger than the control (blue columns). Do some compounds enhance CLAG3 display on the RBC or is it noise?
- In the title of Table 1, can you say specifically what the “CLAG3 export HTS, %” and “NanoLuc HTS, %” mean.
- In Fig 3B, what is the EC50 for? Is it parasite growth? Over how many cycles? Please add to figure legend and to the text.
Author Response
This is an interesting screen where the authors tried to find compounds that blocked the display of CLAG3 on the surface of the parasite infected RBCs. The screen worked by incorporating the HiBit region of Nluc into the region of CLAG displayed on the surface and then using LgBit of Nluc to produce a bioluminescent signal if CLAG3 was displayed. A reduction of signal indicated less CLAG3 display however follow up screens found that the hit compounds where either blocking Nluc activity or reducing the growth of the parasites. In the end no specific inhibitors of CLAG3 exposure on the RBC surface were found but the approach appeared to be robust.
We thank this reviewer for the positive and accurate review of our manuscript.
- Line 225. It says “yellow bars, Fig. 1D” but I can’t see yellow bars in 2D.
Thank you for pointing out this confusing language. According to SigmaPlot (software used to analyze data and create this graphic), this histogram plot used “dark yellow” coloring. We have changed the text to “Cpds, dark yellow histogram bars, Fig. 1D” and hope that readers will use either the “dark yellow” color or the “Cpds” label on the graphic to identify the relevant experimental condition.
2. In Fig 1D, the signal in some wells (bronze columns) is much stronger than the control (blue columns). Do some compounds enhance CLAG3 display on the RBC or is it noise?
This is a good question. We are unclear on what the reviewer is calling “much stronger than the control” and hope to clarify here. This is a histogram of normalized luminescence signals associated with 52,480 compounds, so the height of each bar reflects the number of compounds with a given range of values. For example, the tallest dark yellow bar is centered at x = 99.8 (normalized luminescence) and has a height of 8914 (number of wells), indicating that 8,914 compounds produced luminescence within the range of this bar and equal to the mean luminescence of the “no cpd ctrl”, represented with blue bars. Thus, the height of the bar reflects the number of compounds not the intensity of the luminescence signal. The reviewer may instead be referring to a small number of compounds with higher normalized luminescence values (2 compounds in a dark yellow bar at x = 154.6 and 1 compound at x = 272.4). These three compounds indeed produced increased intensity relative to the controls (in blue). Although we did not purchase these three compounds for secondary studies as our goal was to identify inhibitors of export, our prior studies with chemicals that stimulate the NanoLuc signal suggests that these compounds are typically false positive hits that work by increasing luminescence through interaction with NanoLuc. To clarify this point, we have added “Three compounds producing increased luminescence values > 150% were not evaluated further; prior studies suggest such agonists are false-positives that interact with NanoLuc to produce increased luminescence.” to the legend (lines 206-209).
3. In the title of Table 1, can you say specifically what the “CLAG3 export HTS, %” and “NanoLuc HTS, %” mean.
We have revised the title of Table 1 to clarify that these “Values represent normalized luminescence for each compound, as measured in the indicated high-throughput screen (HTS).”
4. In Fig 3B, what is the EC50 for? Is it parasite growth? Over how many cycles? Please add to figure legend and to the text.
As is conventional in the malaria drug discovery literature, we use “EC50” to refer to “effective concentration” that produces 50% block in a biochemical assay, which is the CLAG3 export luminescence assay in our case. This contrasts with “IC50”, which is generally used to refer to “inhibitory concentration” producing 50% parasite killing in growth assays. We have revised the legend to “Mean ± S.E.M. effective concentration producing 50% reduction in luminescence (EC50) for each hit, determined from dose response experiments using the CLAG3 export assay. Most hits exhibit low micromolar potency in this 24 h assay.” Lines 310-312. We also changed the main text to read “Dose response studies revealed that effective concentrations required to reduce the luminescence signal in the CLAG export assay (EC50 values) were generally in the low micromolar range with several compounds having submicromolar potency (Fig. 3B).” Lines 305-308.
Reviewer 2 Report
Comments and Suggestions for Authors
Shao et al. report on a study demonstrating the use of high-throughput inhibitor screening to identify compounds targeting CLAG3 export and insertion into the infected erythrocyte membrane.
1. Lines149-155: What was the parasite stage at the time of treatment with compounds and when parasites were collected for lysis at 72 h? (See Figure 5). Include Giemsa stained parasite images to show the morphology of the cells following treatment. Synchronized ring stage parasites were maintained for 72h? When were the compounds added to the culture?
2. Section 3.2: What other surface exposed molecules on the ring-infected erythrocyte surface were affected by NanoLuc inhibition? Provide clarification in light of the "false positives" obtained (line 234) and for the reduction in PSAC activity. It appears the target of the compounds is not singularly to CLAG3 but to other components required for PSAC formation and function.
3. Clarify the goal of the screening strategy to show if a broader target(s) involving molecules other than CLAG3 required for PSAC formation will be more promising therapeutically. Perhaps the conclusion can be revised to offer this clarification.
Line 47: "...CLAG3 is conserved all examined malaria parasite species..."
Lines 155-156: "...saponin at pH 7.5. dilution. After a 60 min incubation..."
Line 424: delete "Authors"
Author Response
Shao et al. report on a study demonstrating the use of high-throughput inhibitor screening to identify compounds targeting CLAG3 export and insertion into the infected erythrocyte membrane.
Thank you for your positive comment and your thoughtful review. We have revised the manuscript to clarify and improve presentation.
- Lines149-155: What was the parasite stage at the time of treatment with compounds and when parasites were collected for lysis at 72 h? (See Figure 5). Include Giemsa stained parasite images to show the morphology of the cells following treatment. Synchronized ring stage parasites were maintained for 72h? When were the compounds added to the culture?
Thank you for this question. Our growth inhibition studies used a standardized growth inhibition assay accepted by most malaria drug discovery laboratories, as described in Ref. #35 of the paper (cited on line 149). To clarify the question asked here, we revised the text to state “Ring-stage cultures were seeded with inhibitors after sorbitol synchronization at 0.8% parasitemia and 2.5% hematocrit ….” Lines 149-152. As the 72 h incubation corresponds to 1.5 parasite cycles, control wells without inhibitor generally reach the mid-trophozoite stage. Microscopic imaging as proposed by the reviewer is useful because cell morphologies can provide insights into mechanism of growth inhibition; these are most useful at relatively earlier time points because a long 72 h incubation with compounds that inhibit parasite developmental progression typically lead to killing and attrition of parasites, so workers do not typically perform or include Giemsa-stained images after 72 h. Shorter incubations such as the 24 h incubation with inhibitors used in Figure 4 are more informative. The BF images there show that ART and several of our hits lead to shrunken parasite morphologies, providing insight into their toxicity against parasite maturation.
- Section 3.2: What other surface exposed molecules on the ring-infected erythrocyte surface were affected by NanoLuc inhibition? Provide clarification in light of the "false positives" obtained (line 234) and for the reduction in PSAC activity. It appears the target of the compounds is not singularly to CLAG3 but to other components required for PSAC formation and function.
Section 3.2 describes a counterscreen for compounds that inhibit NanoLuc without affecting the export of CLAG3 or export of any proteins on infected cells. As is conventional in drug discovery programs, we refer to these compounds as “false positives” because they do not block the activity under study (protein export in our case), but rather produce inhibition of the luminescence signal by an unwanted mechanism (blocking NanoLuc-mediated luminescence only). As stated in the Results, “Such false positives may work by blocking LgBit-HiBit association or inhibiting furimazine oxidation to furimamide, the chemical reaction that yields the light signal. NanoLuc structure-function studies support this concern as they reveal a surface binding site that can inhibit substrate access through negative allostery [38]. We therefore devised a cell-based counterscreen to filter our screening results for false positives that act only on the NanoLuc reporter.” Lines 240-245. Thus, these compounds do not interfere with any parasite protein’s export to the host cell surface.
- Clarify the goal of the screening strategy to show if a broader target(s) involving molecules other than CLAG3 required for PSAC formation will be more promising therapeutically. Perhaps the conclusion can be revised to offer this clarification.
This is an excellent suggestion. We have revised the Discussion section to highlight and expand upon this point. We have added a new paragraph “Notably, the split NanoLuc-based RISE assay can be applied to any surface-exposed parasite antigen as it requires only the introduction of a small HiBit tag into an exposed site on the specific parasite antigen of interest. We focused our study on CLAG3 because this antigen has an established role in essential nutrient acquisition an is highly conserved in Plasmodium spp. Inhibition of other antigen’s export may eventually prove to be more promising therapeutically.” Lines 489-494.
Line 47: "...CLAG3 is conserved all examined malaria parasite species..."
Thank you for pointing out this typo. We have corrected it to “CLAG3 is conserved in all examined malaria parasite species”.
Lines 155-156: "...saponin at pH 7.5. dilution. After a 60 min incubation..."
Thank you for pointing out this typo. We have removed “dilution.” to correct this error.
Line 424: delete "Authors"
Thank you for pointing out this typo. We have corrected it.
Round 2
Reviewer 2 Report
Comments and Suggestions for Authors
Line 481: "...nutrient acquisition an is highly conserved..."
Lines 506-510: The conclusion needs to include the utility and immediate relevance of the data obtained in the current study.
Author Response
1. Line 481: "...nutrient acquisition an is highly conserved..."
Thank you for pointing this out. Corrected to "acquisition and is highly"
2. Lines 506-510: The conclusion needs to include the utility and immediate relevance of the data obtained in the current study.
Thank you. We have added "We also identified PEI-1 as a compound that rapidly kills ring stage parasites to reduce CLAG3 export in our screen. It’s IC50 in standard growth inhibition assays, 23 nM, suggests it should be advanced as an antimalarial drug lead, extending findings from prior studies with this chemical [56]." and included a new reference (ref. # 56) for the immediate utility of a screening hit from our work. Lines 510-513.